# Confident Body, Confident Child: Outcomes for Children of Parents Receiving a Universal Parenting Program to Promote Healthful Eating Patterns and Positive Body Image in Their Pre-Schoolers—An Exploratory RCT Extension

**DOI:** 10.3390/ijerph17030891

**Published:** 2020-01-31

**Authors:** Katherine E. Hill, Laura M. Hart, Susan J. Paxton

**Affiliations:** 1School of Psychology and Public Health, College of Science, Health and Engineering, Level 4, George Singer Building, Melbourne Campus, Kingsbury Drive, Bundoora, Victoria 3086, Australia; katherine.hill10@gmail.com (K.E.H.); Susan.Paxton@latrobe.edu.au (S.J.P.); 2Centre for Mental Health, Melbourne School of Population and Global Health, University of Melbourne, Level 4, 207 Bouverie St, Victoria 3010, Australia

**Keywords:** social environment, eating patterns, body image, weight bias, prevention, parents

## Abstract

*Objective*: A four-arm randomized controlled trial (RCT) conducted in Victoria, Australia, previously evaluated parent-report outcomes following Confident Body, Confident Child: a program for parents to promote healthful eating patterns and positive body image in pre-schoolers. This exploratory study evaluated data from children of parents in the trial at 18 months follow-up. *Method*: Participants were 89 children (58 girls, 31 boys) of parents across all RCT arms (group A: Confident Body, Confident Child (CBCC) resource + workshop, n = 27; group B: CBCC resource only, n = 26; group C: nutrition booklet, n = 18; group D: wait-list control, n = 18). Children’s eating patterns, body image and weight bias were assessed via play-based interview. *Results*: Children of CBCC parents reported higher body esteem. Children of nutrition booklet parents reported stronger weight bias. Children of CBCC workshop parents reported lower External Eating. *Discussion*: This exploratory study suggests that CBCC may promote healthy eating patterns and child body image 18 months after parents receive the intervention.

## 1. Introduction

Research examining body satisfaction and unhealthy eating patterns, as well as the related issue of weight bias, increasingly suggests that the foundations of these problems are established in early childhood [1,2,3]. Consequently, identifying effective strategies to protect against these problems, early in life, would be extremely valuable. In this research, we explored eating patterns, body image and weight bias in children whose parents had or had not received the Confident Body, Confident Child (CBCC) resource in a four-arm randomized controlled trial. The initial trial was designed to gather data from parents only to establish program efficacy in changing parental knowledge, attitudes and behaviors. Given that the measures of parents at 6 weeks post-intervention [4] and 6 months and 12 months follow-up [5] showed that the CBCC program was having positive results on parenting, the research team decided to explore whether these parental changes were associated with significant group differences among the children. Hence, a post-hoc study was designed to gather data from children via play-based interview, 18 months after parents received their baseline measurement.

Parents clearly play an important role in shaping the development of unhealthy eating patterns, body dissatisfaction and weight bias in young children [6]. Despite this, many parents of preschool children report knowing little about child body image and acknowledge that they lack strategies for promoting healthful eating, such as using positive feeding practices and encouraging intuitive eating strategies [7]. To fill this gap, a parenting resource, Confident Body, Confident Child (CBCC), was developed as a universal prevention intervention to support and guide parents of children aged 2 to 6 years, to foster healthful eating patterns and positive body image [4]. CBCC includes parent booklets, a poster, a children’s book, a website and a single-session workshop that were developed from research evidence [6,8], informed by consultation with parents [7], and designed to activate behavior change by using theory and evidence from health behavior research [9]. CBCC is a fully manualized resource and delivered by trained health professionals. The program encourages parents to take a weight-neutral approach [10] in encouraging healthful eating patterns and regular physical activity, to avoid fat-talk and appearance-based teasing, to engage in activities that help children develop their self-esteem and sense of accomplishment, and to appreciate bodily functions and abilities not related to appearance (see www.confidentbody.net).

To evaluate CBCC, a randomised controlled trial (RCT) conducted in Victoria, Australia, examined outcomes from parents across four groups: (A) the printed CBCC resource pack and a 2 h face-to-face parent workshop (CBCC resource + workshop); (B) the CBCC printed resource pack only (CBCC resource only); (C) a widely available printed nutrition booklet *Happy Healthy Kids for Life* used as an active control (nutrition control) [11]; and (D) a wait-list control receiving all materials after completion of the final questionnaire (wait-list control). Parents were assessed on their knowledge of parenting risk and protective factors for body image and eating concerns (The Knowledge Test for Body image and Eating patterns in Childhood, [12]). They were also assessed on their positive and negative parenting intentions in relation to behaviors that may influence body image and eating attitudes (Parenting Intentions for Body image and Eating patterns in Childhood, [12]), their feeding practices [13] and family meal time frequency, screen-time use, atmosphere and schedules [4,14].

At 6 weeks post-intervention, the CBCC resource was associated with significant reductions in parents’ intentions to use behaviors that increase the risk of negative body attitudes or unhealthy eating in their children, in parents’ use of feeding practices associated with unhealthy child eating patterns, and in television watching during family meals. Significant increases in parents’ intentions to use positive behaviors and knowledge of child body image and healthy eating patterns were also found [4]. At 6 and 12 months follow-up, the parents who received the CBCC intervention were still showing significantly better scores on measures of knowledge, parenting intentions, and the parental feeding practice of weight restriction, compared to parents who had received the active control nutrition booklet [5]. While this evaluation indicated that CBCC had a positive impact on parenting, it only gathered indirect (parent-report) data on child outcomes and was not designed to assess child outcomes directly. To understand whether the positive impact that CBCC was having on parents had, in turn, the intended impact on children’s body image, weight bias and eating patterns, data from children of the parents involved in the trial also needed examination.

The aim of the current research was to conduct an exploratory follow-up study assessing whether there were significant differences among children of the parents involved in the original CBCC parent trial on measures of eating patterns, child body image and weight bias. Child-reported data were collected directly from a sample of children whose parents participated in the original study using a play-based interview. It was expected that children of parents who received the intervention resources (the CBCC workshop or printed resources) would show greater body satisfaction than children of parents receiving the *Happy Healthy Kids for Life* [11] nutrition control booklet, or the delayed-intervention control. However, because the nutrition booklet provided information to parents on how to foster healthy eating patterns, it was expected that children of parents in this group would report similar improvements to children of parents in the CBCC groups, but significantly better scores than children of parents in the delay-intervention control group, on eating pattern measures. A measure of child weight bias was included as an exploratory endeavour as weight bias had not been previously measured in parents but was considered an important antecedent to child eating and body image concerns.

## 2. Materials and Methods

### 2.1. Participants and Recruitment

In the original study, parents were considered to have retired from participation if they did not respond to three once-weekly requests to complete a measurement occasion. Therefore, if a parent did not complete their 6 week post-intervention questionnaire, they were not invited to complete the next measurement at 6 months. In the current study, only participants who completed all measurement occasions up to and including 12 months follow-up were invited to participate. The eligible sample included 252 parents (n = 58 CBCC resource + workshop, n = 77 CBCC resource only, n = 61 nutrition control, n = 56 waitlist control) who were reporting on 157 girls aged 4–6 years (62%) and 95 boys (38% boys) aged 4–6 years. Of those eligible to participate, 115 (46%) parents consented to their child being interviewed, although only 89 children completed the measures with sufficient data to be included in the current study. Appendix A provides more details on parent characteristics across intervention groups and comparisons between eligible non-completers and those who completed the 18 month follow-up assessment with their child, whose data were included in the current analyses.

Participants were 58 girls and 31 boys of parents from all four groups of the established RCT cohort (CBCC resource + workshop n = 27, CBCC resource only n = 26, nutrition control n = 18 and waitlist control n = 18). For families in the two control conditions, participation in this follow-up study required a 6 month further delay in access to the CBCC resources and this may have contributed to lower participation among the control groups.

### 2.2. Measures

Child eating patterns. An age-adapted version of the Dutch Eating Behavior Questionnaire [15] was used to assess children’s External, Emotional and Restrained Eating patterns. This included a simplified sentence structure and a reduced response set, similar to those used by Carper et al. [16] and van Strien and Oosterveld [15], but did include all items from the original 33 item questionnaire (10 Restrained Eating, 13 Emotional Eating, and 10 External Eating items). Items required children to respond with ‘no’ (1), ‘sometimes’ (2) or ‘yes’ (3). Scores were created by calculating the prorated mean response for items and were derived for children who had responded to at least 70% of the items. Higher scores indicated higher levels of the unhealthy eating pattern. Cronbach’s alpha was 0.74 for External Eating, 0.79 for Emotional Eating and 0.68 for Restrained Eating.

Child body image. Body satisfaction was assessed using an age-adapted version of the Body Esteem Scale [17]. The modified scale comprised 19 questions including “Do you wish you were thinner?” and “Do you feel proud of your body?” The original scale comprised 24 items but in accordance with previous research validating a modified version with children under 7 years old [18,19], the scale was simplified for our sample. Children responded using the simplified response format of ‘no’ (1), ‘unsure’ (2) or ‘yes’ (3). Scores were created by calculating the prorated mean response for items and were derived for children who had responded to at least 70% of the items. Higher scores indicated greater body satisfaction. Satisfactory internal consistency was observed in the current sample (α = 0.76).

Child weight bias. To assess weight bias, children were presented with nine gender-matched silhouette figures ranging in size from very thin (Silhouette 1) to very large (Silhouette 9) (Tiggemann and Pennington, 1990). Similar to Birbeck and Drummond’s [20] procedure, children were asked to select a figure from the array in response to five questions examining children’s perceptions of any social exclusion associated with figure size (including “which figure would you not invite to your birthday party?”, “which figure has no friends to play with?”, “which figure plays all by him/herself?”, “which figure do other children not really like?” and “which figure gets teased by other children?”). The child’s figure selection for each question (1–9) was recorded. The mean score for selected body figures was calculated, where children responded to at least 70% of the items; scores were prorated for the number of items responded to. Higher scores indicated children perceived stronger negative social consequences for larger figures (i.e., greater weight bias). In the current study, Cronbach’s alpha was 0.75.

Body mass index (BMI). A BMI-for-age *z*-score (BMIz) was calculated for children based on the height and weight measurements taken by the first author, using software developed by the World Health Organisation (“WHO Anthro”, 2011, “WHO AnthroPlus”, 2007). Weight status for parents and children was calculated according to the criteria of the World Health Organization [21,22] and Cole et al. [23].

### 2.3. Procedures

Play-based interviews of approximately 30 min were conducted individually with each child participant. To ensure child comfort, parents were offered the choice of conducting the interview either at their home or the University Psychology Clinic, with the parent present. Parents were requested to quietly observe the interviews or engage in other tasks throughout the interview. Families received an AU$20 shopping voucher following participation.

### 2.4. Statistics

To investigate group differences, analysis of covariance (ANCOVA) was conducted with child age and BMIz entered as covariates. Based on the previous trial reporting on parent outcomes, it was expected that: (1) children of parents receiving any type of resource would perform better on all outcome measures (groups A, B, C) than children of parents in the wait-list control (group D); (2) children of parents receiving any CBCC resources (groups A, B) would show greater improvements on outcome measures relating to child body image, than children of parents receiving the nutrition-control (group C); (3) children of parents receiving the CBCC resource + workshop (group A) would show greater improvements on all measures than children of parents receiving just the CBCC printed resource (group B). Hence, planned orthogonal contrasts were used, with the standard difference method for the Weight Bias, Restrained and External Eating scales. As the Emotional Eating scale was non-normally distributed, a Kruskal–Wallis test was conducted to examine between group differences.

The sample of parents in the current study was compared to the sample of parents completing the 12 month measures in the original study who declined participation at 18 months, as this was the only sample from which eligible participants could be drawn. One-way between-groups ANOVAs indicated there were no significant differences on key demographic variables at the 12 month follow-up between participants who completed the current study and those who declined to participate. These findings suggest that the current 18 month sample was representative of the sample at 12 month follow-up.

## 3. Results

### 3.1. Participant Characteristics

Table 1 presents descriptive statistics for age, BMI and child outcome variables for children in each group. Boys were aged between 4.00 and 7.67 years (*M* = 5.88, *SD* = 1.07) and girls between 3.50 and 8.00 years (*M* = 5.63, *SD* = 0.96). Of children, 12% were at higher than average weight. Overall, the proportion of children in the higher weight categories was lower than that obtained in other studies [24]. A one-way between-groups ANOVA indicated that there were no differences between intervention groups on child age (*F* (3, 87) = 1.77, *p* = 0.16) or BMIz (*F* (3, 86) = 1.47, *p =* 0.23). Children’s parents (89 mothers and two fathers) were aged between 30 and 49 years (*M* = 39.09, *SD* = 3.90), with a mean BMI of 25.17 kg/m^2^.

Children’s self-reported External Eating mean scores indicated that children were engaging in these behaviors ‘sometimes’ while they reported ‘no’ to ‘sometimes’ for Emotional and Restrained Eating. This is comparable to data reported by Carper et al. [16] and Damiano, Paxton et al. [25] with samples of 5-year-old girls. On average, children reported high body satisfaction comparable with data reported by Davison et al. [26]. Children generally selected larger figures in response to questions relating to social exclusion and unpopularity, indicating weight bias.

### 3.2. Child Outcome Measures

Results from orthogonal contrasts are shown in Table 2.

Child eating patterns. No differences were observed between groups for Emotional Eating (χ^2^ (3, *n* = 89) = 1.42, *p* = 0.70). Contrasts revealed no differences between groups for Restrained Eating. For External Eating, contrasts revealed no significant group differences between children of parents receiving any resources (groups A, B and C) and children of parents in the delayed intervention control (group D). There were also no significant differences found between children of parents across the two CBCC groups (A and B) compared to children of parents receiving the nutrition control (group C). Analyses comparing the results of the two CBCC groups, however, revealed that children of parents receiving the CBCC resource + workshop (group A) reported lower levels of External Eating than children of parents receiving the CBCC resource only (group B).

Child body image. Planned orthogonal contrasts comparing the outcomes of children of parents who received the CBCC resource across either the workshop or resource-only conditions (groups A and B), to outcomes of children of parents who received the nutrition control resource (group C) revealed that children of parents receiving CBCC reported significantly greater body satisfaction. Children of parents receiving the CBCC resource only (B) reported significantly greater body satisfaction than those receiving the CBCC resource + workshop (A).

Child weight bias. Contrasts revealed that children of parents receiving the nutrition control resource (group C) reported significantly greater weight bias than children of parents who received the CBCC resources either via the workshop or via the resource-only conditions (groups A and B). No differences were observed between children of those receiving any resources (groups A, B and C) and children of those in the delayed intervention control (group D) or between the two CBCC groups themselves (A vs. B).

## 4. Discussion

The current study investigated whether the universal CBCC parenting intervention may have been associated with group differences on child-reported measures of eating patterns, body image and weight bias. We collected child-reported data from a subset of children from parents in the original CBCC evaluation trial as an exploratory extension to the original RCT.

Overall, children reported relatively low levels of Emotional and Restrained Eating, but moderate engagement in External Eating, comparable with previous research [16,25]. No significant differences were found between parent intervention groups on levels of child Emotional or Restrained Eating, as reported directly by children through the play-based interview. However, analyses comparing child External Eating revealed that children of parents receiving the CBCC resource and workshop reported lower levels of External Eating than children of parents receiving the CBCC resource alone. This effect is unexpected, though may be explained by the fact that during the CBCC workshop, despite being presented with material on both healthy body image and eating, parents showed a preference for and greater focus on eating patterns, than the material on body image. It was noted by program facilitators that parents in the workshops often had a stronger urge to discuss their children’s eating patterns than their children’s body image. This engagement may not have been as apparent in the parents receiving the resource only. Nonetheless, it is a very positive finding that children report lower External Eating possibly because their parent attended the CBCC workshop, and may enjoy lower risks for a number of adverse health outcomes into the future, including binge eating [3] and low self-esteem [27,28].

Children of parents receiving any CBCC resources reported higher levels of body satisfaction than children of parents receiving the nutrition resource only. This finding is particularly encouraging, although it was associated with a small effect size and therefore warrants cautious interpretation. This finding suggests that the CBCC intervention may have had a positive impact on children’s body satisfaction, even though there was no researcher contact with children until 18 months post-intervention with parents, and even in the context of an already high level of body satisfaction. This is important given that body dissatisfaction has been prospectively associated with maladaptive eating attitudes and dieting, across childhood [29].

Interestingly, a larger benefit was seen for children of parents receiving the CBCC resource only than for those receiving the CBCC resource + workshop (significant difference with medium effect size), suggesting that the delivery method of the resource may have had a differential effect on children’s body image. At 6 week follow-up in the original RCT evaluation, parents who received the CBCC resource only reported having read significantly more of the printed CBCC resource pack than parents who received the CBCC resource + workshop [4]. In addition, the resource-only group also reported having used it more with their child (i.e., reading the included children’s picture book *Shapesville* [30]) compared to parents who received the CBCC resource + workshop [4]. This book was chosen for inclusion in our parent resource pack as there was existing research with school-age children showing it had positive effects on body satisfaction [31]. It is therefore possible that the greater engagement with the printed resources may have led to better body image outcomes for the children. Another possibility is that reading the printed resources was indicative of greater implementation of CBCC recommendations generally. A significant limitation of the current study, however, is the absence of baseline measures to ascertain that any significant differences across groups resulted after the intervention and were not apparent at randomisation. In addition, the sample size was small and underpowered for the analyses undertaken. However, as a pilot investigation, the positive outcomes reported by children at 18 months post-intervention in this study, suggest that further research using a pre-post design with child participants, is certainly warranted.

Children in the sample perceived stronger negative social consequences for larger figures, suggesting an awareness of weight bias that is consistent with stereotypical societal ideals and previous research [32,33,34,35,36]. Analysis of child scores across parent intervention groups indicated that children of parents receiving the nutrition resource reported higher levels of weight bias than children of parents in the CBCC intervention groups. Although this significant finding was in the hypothesised direction, these results suggest that the nutrition resource may be leading to a strengthening of children’s perceived negative outcomes for larger bodies, rather than CBCC leading to positive or protective outcomes. Indeed, in response to questions relating to social exclusion and unpopularity, on average, children of parents receiving the nutrition resource selected one whole figure size larger (i.e., Silhouette 7) on the 9-figure rating scale than children of parents in the other groups (A, B and D; Silhouette 6). This difference occurred despite the nutrition information being provided to *parents only*, perhaps indicating that parents are translating the nutrition information to their children in ways that increase negative messages about weight. However, as noted above, without baseline child measures, we cannot be certain that any significant differences across groups were indeed a direct effect of the intervention received. Nutritional resources, like *Happy Healthy Kids for Life*, are easily accessible, low-cost and often distributed by health professionals. If nutrition information in the absence of parental guidance on how to foster positive body image does indeed lead to an increase in weight bias in children, then easy access and frequent distribution of nutrition resources may perpetuate weight bias at a societal level. Further research examining the effects of nutrition education on weight bias in children is certainly therefore warranted.

A strength of this study is its focus on salient measures of eating patterns, body image and weight bias, across both boys and girls in the preschool years, a novel frontier for nutrition and body image research. Further research is required to understand how child reported data changes overtime from baseline to post-intervention and follow-up, after parents receive the CBCC program. Given the limitations of the current sample, including that children were from educated families with lower than average proportions in the higher BMI range, more work is also required to understand how the program impacts on parents and children from diverse backgrounds and lived experience of an eating disorder. Understanding whether the CBCC program impacts boys and girls differently is also an important goal, as our sample was too small to analyze effect modification by gender (the spread of boys across the four intervention groups was insufficient; Group D n = 5 boys). However, if the findings of the current study are replicated and indeed found to be the result of program delivery, then the CBCC resource has the potential to significantly and positively impact on child health outcomes and reduce risk for unhealthy eating patterns, body dissatisfaction, and the negative implications with which these phenomena are associated.

## 5. Conclusions

CBCC appears to be associated with a number of significant outcomes in children at 18 months follow-up, despite the intervention being delivered to parents only. Promising findings include lower External Eating and significantly greater body satisfaction in children of parents receiving the CBCC resource. Of concern is the finding that providing parents with nutritional information in the absence of strategies to foster positive body image could possibly lead to an increase in child weight bias, though further research is required to ascertain the causal relationship between the two.

## Figures and Tables

**Table 1 ijerph-17-00891-t001:** Descriptive statistics for age, BMI and child outcome variables.

Variable	Group A		Group B		Group C		Group D		Total	
	*M*	*SD*	*M*	*SD*	*M*	*SD*	*M*	*SD*	*M*	*SD*
Child BMIz	0.23	0.89	0.04	0.79	0.50	0.97	0.50	0.81	0.29	0.87
Parent BMI	27.53	6.83	23.73	4.21	24.10	4.02	24.57	4.67	25.17	5.40
Child Age	5.41	0.98	5.96	0.92	5.92	0.94	5.58	1.12	5.71	1.00
Parent Age	38.52	3.96	39.54	3.68	39.81	3.43	38.67	4.64	39.09	3.92
Body Satisfaction ^a^	2.61	0.04	2.78	0.04	2.55	0.05	2.59	0.27	2.65	0.28
Weight Bias	6.38	1.40	6.29	1.70	7.28	1.42	6.17	1.73	6.49	1.59
External Eating	1.77	0.47	2.11	0.43	2.07	0.41	2.01	0.55	1.99	0.48
Emotional Eating ^b^	1.39		1.26		1.38		1.35		1.23	
Restrained Eating	1.48	0.42	1.32	0.28	1.46	0.44	1.46	0.42	1.42	0.39

CBCC resource and workshop (group A); CBCC resource (group B); nutrition resource (group C) and; delayed intervention control (group D). ^a^ Adjusted means and standard error are reported. ^b^ Median values are reported due to non-normal distribution of scores.

**Table 2 ijerph-17-00891-t002:** Results of planned orthogonal contrasts on child outcome variables.

Contrasts		Difference (Estimate-Hypothesised)	SE	*p*	*η* ^2^
Body Satisfaction					
Group D	Groups C, B and A	−0.01	0.02	0.48	
Group C	Groups B and A	−0.04	0.02	0.03 *	0.05
Group B	Group A	0.06	0.02	0.01 *	0.08
Weight Bias					
Group D	Groups C, B and A	−1.43	1.33	0.29	
Group C	Groups B and A	1.88	0.94	0.048 *	0.05
Group B	Group A	−0.10	0.47	0.84	
External Eating					
Group D	Groups C, B and A	0.05	0.13	0.71	
Group C	Groups B and A	0.13	0.09	0.15	
Group B	Group A	0.16	0.05	0.002 **	0.11
Restrained Eating					
Group D	Groups C, B and A	0.04	0.13	0.76	
Group C	Groups B and A	0.05	0.09	0.59	
Group B	Group A	−0.06	0.05	0.20	

*η*^2^ effect size: 0.01 = small, 0.09 = medium, and 0.25 = large. * *p* < 0.05; ** *p* < 0.01.

## Data Availability

The datasets used and analysed during the current study are available from the corresponding author upon reasonable request.

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
