# Peer review of "Confident Body, Confident Child: Outcomes for Children of Parents Receiving a Universal Parenting Program to Promote Healthful Eating Patterns and Positive Body Image in Their Pre-Schoolers—An Exploratory RCT Extension"

_ijerph, 2020, doi:10.3390/ijerph17030891_

Round 1

Reviewer 1 Report

Comments have been addressed by the authors and I would recommend the manuscript for publication 

Author Response

We thank the reviewer for their consideration of our revised manuscript and look forward to seeing the paper published in this journal.

Reviewer 2 Report

The revision appear appropriate. This manuscript appears ready for publication.

Author Response

(The authors gave the same response as above.)

Reviewer 3 Report

This manuscript has addressed most of my previous concerns. As I wrote in my previous review, I do understand the reason for not performing a gender stratified analysis (small sample). I would, however, have expected the authors to add a sentence about this in the discussion. And, I don't understand the reason for not performing an interaction analysis on the impact of gender for the outcomes, this analysis can often be meaningful even if the sample is not large enough for a stratified analysis.

Author Response

We thank the reviewer for taking the time to comment on our resubmission. We have chosen not to provide a gender analysis in this paper for the following reasons:

Although we agree that examining differences between boys and girls would be interesting and should be a focus of body image and prevention research, examining gender effects was not an aim of this study. We did not recruit equal numbers of boys and girls as our sampling was post-hoc through parents engaged in the original RCT, hence analyses assessing gender differences may appear to be ‘data mining’ to the paper audience, as these were not intended in the design of the current study or the original RCT. As a ‘brief report’ we believe that it would unnecessarily extend the paper to include information on gender interaction analyses. Further, we have conducted a further RCT with data gathered at both baseline and follow-up, with fewer intervention groups, which is better suited to these kinds of analyses. The sample of boys is too small to look at how gender might impact on effects across groups. We have a total of only 5 boys in Group D, and 7 boys in each of Groups B and C. In addition to being post-hoc analyses any statistical significance testing will be highly prone to inflated error and any outcomes would need to be interpreted with extreme caution.

Group A

Group B

Group C

Group D

Total

Male

11

8

7

5

31

Female

16

18

11

13

58

We have now included the following sentence on p.9 of the manuscript (rows 307-310) to note why we did not analyse gender:

Understanding whether the CBCC program impacts boys and girls differently is also an important goal for future research, as our sample was too small to analyze effect modification by gender (the spread of boys across the four different intervention groups was insufficient; Group D n=5 boys).

This manuscript is a resubmission of an earlier submission. The following is a list of the peer review reports and author responses from that submission.

Round 1

Reviewer 1 Report

Many thanks for the very interesting manuscript submitted for a communication. Building on the previous CBCC study this short study fills a gap in the previous work, and as it was conducted 18 months post intervention provides a a good insight as to the long term impact of such an intervention on the target audience. 

Although the sample size is small and subject to some bias the authors have acknowledged these in their discussion, and have clearly outlined the potential limitations of their work. However the results warrant publication and support a growing area of research. 

My comments are very minor:

At no point in the manuscript do the authors describe the country/area or context in which this study took place. I would recommend this is highlighted in the abstract and main text The table formatting on both tables need to be addressed which may have resulted from the change to PDF. Headings and numbers are out of line and confusing to interpret. These need to be addressed by the authors or editors.  Under procedures L158 I would recommend that the authors make clear that each interview was conducted with a child individually or in a group. The inference is that they were individual but I feel this shoudl be explicit. 

Author Response

1. The following text has now been added to the first line of the Abstract (p1) and the Introduction (line 67, p2) to clarify where this study took place: ‘…conducted in Victoria, Australia’

2. We apologise for the formatting errors; these seem to have occurred on submission. We have now tried to rectify this on the re-submitted manuscript draft and hope that it does not arise again.

3. We have now noted that interviews were conducted as individual interviews by adding a note on line 160, p4.

Reviewer 2 Report

“Confident Body, Confident Child: outcomes for children of parents receiving a universal parenting program to promote healthful eating patterns and positive body image in their pre-schoolers – an exploratory RCT extension” examined child outcomes associated with a parenting program designed to improve healthy eating and positive body image. The RCT had four conditions with about 23 participants per condition: those who received the CBCC paper resources and a single-session workshop, those who received CBCC paper resources, those who received widely available nutrition booklet, and a passive control condition. Children were assessed 18 months post-intervention. Overall, there were very few differences in child eating patterns, child body image and child weight bias (such that group A had less external eating than group B; that groups AB had greater body satisfaction than group C, that Group B had greater body satisfaction than group A and that group C had more weight bias than A and B).  

The authors present their results clearly and effectively demonstrate the need for early intervention in body image and healthy eating patterns. The intervention is also thoughtfully designed and seems comprehensive in its scope. The use of an active and passive control is also an excellent methodological addition, providing nuance to exactly where the variation in child outcomes derives.

The primary limitation of this study is its severely limited power. 23 participants per cell is simply not a large enough sample to draw meaningful conclusions. This problem is compounded by engaging in multiple comparisons across four outcome variables, often further diminishing the sample size. Conducting four significance tests with 91 participants (A vs BCD), four significance tests with 74 participants (AB vs C) and four significance tests with 54 participants (A vs B) increases the likelihood that the researchers are capturing chance variation rather than systematic differences caused by the intervention. The fact that one of the four observed differences was in the unexpected direction, and that another observed difference was attributed to the active control rather than the focal manipulation is consistent with these limitations (which are not acknowledged in the manuscript).

Although the authors are explicit with the exploratory nature of the study, I believe that this manuscript should not be published through IJERPH (in its current form) given the considerable risk both Type 1 and Type 2 error. This exploratory study would serve as a compelling “Study 1” in a multi-study manuscript, as long as an adequately-powered follow-up trial followed it.  

Author Response

1. The authors thank Reviewer 2 for their detailed consideration of our research. We acknowledge that this study does have significant limitations and provides only brief pilot data to motivate further research. As this study reports on outcomes from a previously reported RCT, we believe that this data is best published as a short report in IJERPH rather than a multi-study paper. The original trial has already been published and any new trial with sufficient sample size and experimental design would require significantly different methodological detail to warrant a separate and independent paper. 

2. To address the concerns of Reviewer 2, we have noted the sample size limitation in the Discussion (line 279-282, p.12): ‘In addition, the sample size was small and underpowered for the analyses undertaken. However, as a pilot investigation, the positive outcomes reported by children at 18-months post-intervention in this study, suggest that further research using a pre-post design with child participants, is certainly warranted.’

Reviewer 3 Report

The program for parents to promote healthy eating patterns and positive body image in younger children is an interesting, current, and needed idea/approach.

The study scope of extending the effectiveness of the program to children of parents who have had some form of the program (or wait list) is also quite novel.

I find the article well-written, and well-constructed. And while the absence of pre-test data limits validity some, the exploratory nature of these findings and potential to influence further study is important.

One item of clarification: Is more demographic data available on the study sample? The authors mention that the children came from educated homes, but provide no further sample demographic data (on SES, racial/ethnic background, etc.).

This article is close-to-publish ready and the authors should be commended for their work, which should make a nice addition to this journal.

Author Response

1. The authors are grateful to Reviewer 3 for their thorough consideration of our study. A large amount of detail on the parents in the study has been published in the papers describing the original RCT:

Hart, L.M., S.R. Damiano, and S.J. Paxton, Confident Body, Confident Child: a randomised controlled trial evaluation of a parenting resource for promoting healthy body image and eating patterns in 2- to 6-year old children. International Journal of Eating Disorders, 2016. 49(5): p. 458-472.

Hart, L.M., S.R. Damiano, and S.J. Paxton, Confident Body, Confident Child: evaluation of a universal parenting resource promoting healthy body image and eating patterns in early childhood – 6 and 12 month outcomes from a randomized controlled trial. International Journal of Eating Disorders, 2019. 52(1): p. 1-11.

However, in line with the request for more information on parent participants from Reviewers 3 & 4, we have now provided a supplementary file with a table outlining characteristics of parents.

Reviewer 4 Report

This is a long term evaluation of an intervention in a key child health area, and as such an important study. The authors correctly addresses the main limitation of this study, which is the lack of baseline measures of the child outcomes investigated. This is thus a cross-sectional analysis of the child health outcomes 18 months after exposure ec. If ventiont of the interthe intervention.

There two major issues that the authors to not raise at all which need to be addressed.

Selection into participation is a key issue for all intervention studies, and the cross-sectional design of the evaluation makes this an even more important issue. The authors have looked at selection into participation at the 18 months follow-up in the participants of the 12 months follow-up by looking at "statistically significant" differences in demographic factors. This is problematic in two different ways. First of all the key factors here are not demographic but the effects of the intervention. If the 18 months participants had higher mean intervention effects than the 12 months participants in the parent reported outcomes at 12 months, this could potentially explain all of the positive effects found at 18 months. Further, the authors seem to believe that having no "statistically significant" differences means that the demographic profile was similar in the 12 and 18 months follow-up studies. This is only somewhat true in very large samples. In these small samples it means very little. When looking at similarities between two groups, p-levels are actually reversed, so that a p-level of>0.95 means that they are the same on the 5% level. The implications of this is that the authors need to create a table where 18 and 12 months participants are compared not only for parental demographic characteristics but also for effect sizes in parent reported outcomes at the 12 months study. This table could be added as a supplemental file, and the results discussed in the limitations section. 2/3 of the particpants at the 18 months follow-up were girls and  thus only 1/3 were boys. this is obviously not a random distribution and the crude results therefore need to be presented in a gender stratified manner. Because of the comparatively small number of participants in each arm I understand that the main statistical analysis might have to be made in boys and girls together, but gender differences should be looked for in interaction analyses.  Minor...The layout of Table 1 is very poor in my copy. 

Author Response

1. In response to the request for more information on selection into participation, we have now provided a supplementary document with more detail on comparisons between the parents who were eligible for the current study but did not participate, with those who did complete the 18-month follow-up assessment with their child, which is the focus of this paper. Please see Supplementary File 1 and Table 1 within the document. 

2. Regarding child gender: The distribution of child gender (2/3 girls) was the same at baseline in the original RCT as it is in the current sample at 18-months follow-up (see Table 1, in Supplementary File 1). This seems to be a function of parent interest in learning about body image among girls in particular, rather than any drop-out occurring among parents of boys.

Because of the small cell sizes, already highlighted by Reviewer 2, and the need to minimize multiple comparisons inflating chances of Type-1 error, we are not able to stratify by gender but hope to do so in future studies with larger samples.